

# An exploratory pilot study of mechanisms of action within normative feedback for adult drinkers

Alexis Kuerbis[1], Frederick J. Muench[2], Rufina Lee[1], Juan Pena[1] and Lisa Hail[3]

[1] Silberman School of Social Work, City University of New York, Hunter College, New York, NY, United States
[2] Department of Psychiatry, Northwell Health, Great Neck, NY, United States
[3] School of Psychology, Fairleigh Dickinson University, Metropolitan Campus, Teaneck, NJ, United States

## ABSTRACT

**Background.** Normative feedback (NF), or receiving information about one's drinking compared to peer drinking norms, is one of the most widely used brief interventions for prevention and intervention for hazardous alcohol use. NF has demonstrated predominantly small but significant effect sizes for intention to change and other drinking related outcomes. Identifying mechanisms of action may improve the effectiveness of NF; however, few studies have examined NF's mechanisms of action, particularly among adults.

**Objective.** This study is an exploratory analysis of two theorized mechanisms of NF: discrepancy (specifically personal dissonance—the affective response to feedback) and belief in the accuracy of feedback.

**Method.** Using Amazon's Mechanical Turk, 87 men ($n = 56$) and women ($n = 31$) completed an online survey during which they were asked about their perceptions about their drinking and actual drinking behaviors. Then participants were provided tailored NF and evaluated for their reactions. Severity of discrepancy was measured by the difference between one's estimated percentile ranking of drinking compared to peers and actual percentile ranking. Surprise and worry reported due to the discrepancy were proxies for personal dissonance. Participants were also asked if they believed the feedback and if they had any plans to change their drinking. Mediation analyses were implemented, exploring whether surprise, worry, or belief in the accuracy of feedback mediated severity of discrepancy's impact on plan for change.

**Results.** Among this sample of adult drinkers, severity of discrepancy did not predict plan for change, and personal dissonance did not mediate severity of discrepancy. Severity of discrepancy was mediated by belief in the accuracy of feedback. In addition, viewing one's drinking as a problem prior to feedback and post-NF worry both predicted plan for change independently.

**Conclusions.** Results revealed that NF may not work to create personal dissonance through discrepancy, but belief in the accuracy of feedback may be important. It appears the more one believes the feedback, the more one makes a plan for change, suggesting practitioners should be mindful of how information within feedback is presented. Findings also indicate NF may work by validating a preexisting perception that drinking is a problem instead of creating concern related to discrepancy where none existed. Limitations regarding generalizability are discussed.

Corresponding author
Alexis Kuerbis,
ak1465@hunter.cuny.edu

## INTRODUCTION

Hazardous and/or harmful use of alcohol, such as drinking to excess or in the context of comorbid medical or mental health conditions, is a public health problem that causes economic and social burden across the globe (*World Health Organization.*, *2014*). In the United States in 2013, just under a quarter (24.7%) of adults aged 18 and older reported binge drinking (consuming five or more drinks on one occasion), and 6.8% reported heavy drinking (drinking beyond the recommended guidelines for safe alcohol use given by the National Institute for Alcohol Abuse and Alcoholism (*National Institute on Alcohol Abuse and Alcoholism*, *2013*)) in the past month (*Substance Abuse and Mental Health Services Administration*, *2014*). Many hazardous drinkers will be at risk for serious health consequences but do not perceive their drinking as requiring formal treatment (*Cunningham et al.*, *2001*; *Hester & Delaney*, *1997*; *Rosenberg*, *1993*). Due to the prevalence rates of hazardous drinking, the avoidance of formal treatment, and the recent call for expanded prevention services within health care reform (*Arvantes*, *2010*), there is an increasing need for effective brief interventions that address harmful alcohol use at both the individual and population levels (*McCambridge & Cunningham*, *2014*; *Dotson, Dunn & Bowers*, *2015*) outside of specialty health care settings.

Several brief interventions have been developed (*McCambridge & Cunningham*, *2014*) to prevent alcohol use disorder (AUD) and to reduce alcohol related consequences. Brief interventions tend to be less than an hour long, or sometimes as short as a few minutes, and are usually implemented by a non-specialist, such as a primary care physician, or, in some cases, on a computer. One such brief intervention is normative feedback (NF), a widely used intervention for prevention of and early intervention for hazardous alcohol use (*Cunningham, Murphy & Hendershot*, *2015*). During NF, individuals are assessed and then provided with information about how their drinking patterns compare to drinking norms of their peers of the same gender and age. The purpose of NF is to provide a drinker with a tool for understanding their own, potentially excessive, drinking and ultimately motivate them to reduce their drinking to safe levels.

There are a variety of forms of NF, and it is implemented in a myriad of ways. For example, NF can be implemented as a standalone intervention; as part of more detailed feedback that includes information on health consequences, referred to as personalized feedback (*Walters & Neighbors*, *2005*); or alongside or within another therapeutic intervention, such as Motivational Enhancement Therapy (MET) (*Walters & Neighbors*, *2005*; *Murphy et al.*, *2004*). NF is often provided in community-based settings, such as primary care (*Maisto et al.*, *2001*; *Collins, Carey & Sliwinski*, *2002*), via multiple modalities (e.g., web-based, mailed, in-person) (*Walters & Neighbors*, *2005*; *White*, *2006*; *Walters & Woodhall*, *2003*; *Cunningham et al.*, *2012*; *Cunningham et al.*, *2009*). Given its relative simplicity, it is somewhat surprising that NF demonstrates consistent effectiveness in helping individuals reduce drinking at statistically significant levels across all modalities, contexts (e.g., within therapy or alone), and structures (e.g., a single item or detailed feedback about multiple health indicators) (*Cunningham et al.*, *2001*; *Dotson, Dunn & Bowers*, *2015*; *Walters & Neighbors*, *2005*; *Murphy et al.*, *2004*; *White*, *2006*;

*Cunningham et al.*, *2012*; *Miller, Benefield & Tonigan*, *1993*; *White et al.*, *2007*; *Doumas & Hannah*, *2008*; *Walters et al.*, *2009*; *Kuerbis et al.*, *2014*; *Riper et al.*, *2009*; *Cadigan et al.*, *2015*; *Zimmerman & Fischhoff*, *2014*; *Cunningham et al.*, *2010*).

It is important to note that the preponderance of the above described research on NF has primarily focused on college students and/or their non-college same-age counterparts (*Walters & Neighbors*, *2005*; *Reid & Carey*, *2015*; *Miller et al.*, *2013*). This is in part because of the well documented prevalence of high risk drinking on college campuses and among individuals at this stage of late adolescence (*Substance Abuse and Mental Health Services Administration*, *2014*). Only a few studies have examined the impact of NF among adult drinkers beyond the college age years (e.g., *Cunningham et al.*, *2001*; *Cunningham, Murphy & Hendershot*, *2015*; *Maisto et al.*, *2001*; *Cunningham et al.*, *2012*; *Cunningham et al.*, *2009*; *Kuerbis et al.*, *2014*; *Cunningham et al.*, *2010*). Within those few studies, NF again demonstrates preliminary effectiveness among adults at a spectrum of ages; however, this effectiveness is less well established, especially long term (*Cunningham et al.*, *2010*), than for college-aged individuals.

While NF is considered an effective intervention, particularly for individuals in late adolescence, it can potentially be improved to better facilitate both individual and population health (*McCambridge & Cunningham*, *2014*). While some studies demonstrate impressive effect sizes of NF, most studies demonstrate small effect sizes (*Dotson, Dunn & Bowers*, *2015*; *Cunningham et al.*, *2012*; *Cadigan et al.*, *2015*). There is also evidence that NF does not impact drinking outcomes as much as has been previously suggested (*Foxcroft et al.*, *2015*; *Gaume et al.*, *2014*), particularly outside the context of primary care or for more than six months (*Cunningham et al.*, *2010*).

One way to improve an intervention is to understand its mechanisms of action. If one identifies and understands how an intervention works, one can hone the intervention—improving its efficacy and cost-effectiveness. Although some studies have examined the mechanisms of action of personalized feedback (*McCambridge & Cunningham*, *2014*) and NF (*Neighbors et al.*, *2016*), mechanisms remain relatively unexplored (*Gaume et al.*, *2014*).

## Proposed mechanisms of action of NF

Two potential mechanisms of action of NF explored in the literature are (1) discrepancy, including both external discrepancy (*Neal & Carey*, *2004*), the difference between one's perceived norm and the actual norm, and internal discrepancy (*Neal & Carey*, *2004*), the dissonance between an individual's drinking and an internal standard of comparison, such as one's "ideal" drinking behavior; and (2) perceived accuracy of feedback.

Discrepancy has been explored in a number of ways due to its different forms and definitions. Social Norms Theory (*Perkins*, *2003*) is used to explain the effect of NF—such that once the perception of how much others drink is reduced, then the individual's drinking will be reduced in kind. Some studies support this theory, demonstrating change in perceived norms as a significant mediator of NF in reducing drinking among college students (*Walters et al.*, *2009*; *Neighbors et al.*, *2016*; *Borsari & Carey*, *2000*; *Neighbors, Larimer & Lewis*, *2004*; *Walters, Vader & Harris*, *2007*). Other studies show that while change in perceived norms relates to intentions to change alcohol use among college

students, it does not necessarily lead to actual reduced drinking (*Neal & Carey*, *2004*; *Larimer et al.*, *2007*; *Murphy et al.*, *2010*). Notably, young adults may react to discrepancy differently from older adults due to their particular life stage (*Erikson*, *1968*). For example, a heightened need for peer approval during late adolescence may cause a more potent effect of NF (*Kuerbis et al.*, *2014*). As such, mechanisms of action of NF may differ across age groups.

Within the context of MET (*Miller et al.*, *1999*), NF is thought to develop internal, self-ideal discrepancy (*Murphy et al.*, *2010*; *Miller & Rollnick*, *2002*). In MET, a heightened awareness of discrepancy between one's image of oneself and one's actual behavior is thought to create conflict for the individual, resulting in discomfort with the status quo (*McNally, Palfai & Kahler*, *2005*) and thus enhancing motivation to change. This conflict is described many ways—including affectively (*McNally, Palfai & Kahler*, *2005*)—such as fear, worry, or a neutral uneasiness with the difference between perception and behavior. This conflict is conceptualized here as an affective response to discrepancy and is termed *personal dissonance*. Only two studies thus far directly examined the affective response related to NF and drinking with any population (*Kuerbis et al.*, *2014*; *McNally, Palfai & Kahler*, *2005*). Results of one study demonstrated that among adult problem drinking men-who-have-sex-with-men (MSM), NF within MET elicited concern and worry, but these responses were not related to reduction in post-NF drinking (*Kuerbis et al.*, *2014*). The second study (*McNally, Palfai & Kahler*, *2005*) also found no relationship between developed affective dissonance and drinking outcomes among heavy drinking college students. Given that only one study has examined this process among an adult population beyond college age and only in the context of MET, further research is needed to explore affective responses to NF and their role in behavior change among adults, at a range of ages.

Perceived accuracy of NF (e.g., belief that norms used for drinking comparisons and the results of the evaluation are true or valid) may also mediate NF's efficacy. Decision dilemma theory (DDT) is a behavioral economic theory that posits that poor decision making (e.g., continued hazardous drinking) results from receiving "equivocal" feedback (*Hantula & DeNicolis Bragger*, *1999*; *Bowen*, *1987*) or "feedback for which multiple (positive or negative) interpretations can be constructed" (*Bowen*, *1987*). An individual who perceives NF as highly equivocal might view the information as inaccurate or not applicable to their personal situation. As a result, the individual would dismiss the information, nullifying the effects of NF. In their study of adult drug users who received NF, Amrhein and colleagues (*Amrhein et al.*, *2003*) noted a subgroup of participants who disagreed with the feedback they received (thus viewed it as "highly equivocal"). This subgroup demonstrated more stable or increased drug use post-NF. Were this the mechanism of action of NF, it would be important to modify NF to present information in a "believable way" to maximize its effect. A study of 90 adult problem drinking MSM who received NF specifically tested this and found no support for DDT as an explanatory mechanism for NF (*Kuerbis et al.*, *2014*). A primary problem with this study was that perceived accuracy of feedback was measured by a third party observer. By viewing video-taped psychotherapy sessions in which NF was provided, observers coded statements clients made in session to measure the client's perceived accuracy of the feedback (e.g., "Your data is wrong."). In-session statements

made about this perception were not explicitly solicited by therapists and thus were made inconsistently across participants. Participants were also not directly asked whether they believed the feedback outside of session.

The current study aimed to expand the limited knowledge about the potential mechanisms of action within NF, specifically affective responses to NF and belief in the accuracy of the feedback, by implementing a feasibility and exploratory pilot study of NF on adult drinkers. Given that many individuals with varied drinking patterns are receiving feedback via online mechanisms, such as alcoholscreening.org, we explored whether it was possible to gather data about personal reactions to feedback and test the mechanisms of action of web-based NF with a heterogeneous sample of drinkers. In this feasibility study, personal dissonance, as defined by level of *surprise* and *worry*, and *belief in the accuracy of feedback* were tested as mediators of *severity of discrepancy* (the magnitude of the difference between one's drinking and peer norms) on a drinker's plan to reduce drinking. It was hypothesized that both personal dissonance and belief in feedback would emerge as mediators of severity of discrepancy predicting planning for behavior change.

## METHOD

### Recruitment

Participants were recruited through Amazon Mechanical Turk (MTurk), an online labor market. MTurk is a online platform through which "workers" are contracted to complete tasks, called "human intelligence tasks" (HITs), such as beta testing software or providing consumer opinions. For completing a HIT, workers are compensated by the publisher of the task, called "requesters." Compensation is generally low—often times below $1—and commensurate with the task intensity. Increasingly, MTurk is being used for social sciences research with comparable results to more traditional sampling methods, when validity checks are included in the design (*Mason & Suri*, *2012*; *Muench et al.*, *2014*).

### Eligibility

Participants were workers registered on MTurk. To participate in this HIT, they had to have received a HIT approval rating of 95% out of at least 500 completed HITs by other requesters. This limited the sample to workers deemed acceptable by other requesters and with adequate literacy in computer and internet use—thus providing increased assurance that participants could optimally understand this computer based intervention. Eligibility was limited to those residing in the United States.

### Procedures

Eligible workers viewed the advertisement for the HIT *Answer questions about a health behavior and receive feedback*. Within the HIT, participants were directed to a link to an external web-based survey, hosted by SurveyMonkey.com. The initial page of the web-based survey provided information about the study, and only those who consented to participate clicked through to the actual survey. Within the survey, participants answered the question: "Do you drink alcohol?" Participants who answered no ($N = 38$) were directed to a survey related to exercise and were excluded from this analysis. Those who

responded yes were directed to a survey about alcohol, which included NF about their own use (described further below). At the end of the survey, participants were provided with the reference information for the data used in the feedback. The survey took about 6 min to complete. Once completed, participants were provided with a code for their MTurk account, signaling the requester to review the responses and approve compensation. Participants were compensated 50 cents for survey participation, an average rate of compensation for comparable tasks. Workers were barred from retaking the survey.

## Measures
### Demographics
Participants were asked their age and gender.

### Perceptions of participants' drinking
Participants were asked a number of questions about their perception of their drinking prior to NF. On a scale of 1 (Not at all) to 8 (Extremely), participants were asked if the amount that they drank was excessive (*excessive*) and whether their current *drinking is a problem* for them or others (*problem for others*) in their lives. Finally, drinkers were asked "When you have not planned to drink, how much effort does it take for you not to drink alcohol when it is presented to you?" (*effort*). The effort question attempted to capture level of control over drinking.

### Self-reported quantity and frequency of drinking
Participants were asked to report on their actual drinking through a series of questions based on the Form 90 QFV-30 (*Miller & Del Boca*, *1994*). Responses were used to provide NF, described below.

### Personal dissonance
Immediately after feedback was provided, participants were asked to rate, again on a scale of 1 (Not at all) to 8 (Extremely), (1) how surprised are you by the feedback? (*surprise*) and (2) how much does the feedback worry you? (*worry*). These were used as proxies for personal dissonance resulting from NF.

### Expectation
Participants were asked "Is this feedback, ..?" and then provided with three options: better than expected, as expected and worse than expected.

### Newness of information
Participants were asked if the information provided in the NF was new. They could respond yes or no.

### Plan to change
In the last section of the survey, participants were asked about their plan related to drinking for the next 30 days. This question was categorical and responses ranged from (0) no change in drinking to (3) quitting drinking. Participants were also asked to rate on a scale of 1 (Not at all) to 8 (Extremely) their *commitment* to and *confidence* in their ability to achieve their plan.

### Belief in the accuracy of NF

To avoid biasing item responses earlier in the survey, the last question was "how accurate does the information you received about your drinking seem to you?" A 5-point Likert response scale included anchors –2 Not at all accurate to 2 Definitely accurate. Zero was "Not sure."

### Severity of discrepancy

Participants were asked to rank themselves as to how much they drink compared to their same gendered peers in the form of a percentile rank. *Severity of discrepancy* was calculated as the difference between the participant's initial reported percentile rank and actual percentile rank (calculated based on reported standard drinks per week)—a measure similar to those used in other studies (*Walters et al.*, *2009*; *Walters, Vader & Harris*, *2007*). Higher numbers indicated greater discrepancy. Positive *severity of discrepancy* indicated that participants drank more compared to their peers than they estimated.

## Normative feedback

Normative comparisons were based on the 2009 National Survey on Drug Use and Health (*Substance Abuse and Mental Health Services Administration*, *2010*). Based on participants' reported average number of standard drinks per week, they were directed to one of 18 different possible statements of feedback, each with a unique percentile rank based on gender and quantity of standard drinks per week. Feedback contained two sentences; for example: "You reported drinking 1 standard drink per week. You drink more than 27% of men." While we asked about binge drinking, binge drinking was not a factor in determining feedback, nor were participants educated about binge drinking.

## Analytic plan

Descriptive analyses were performed to evaluate overall participant perceptions of health behaviors, their reactions to NF, and their plans for change. Correlations between the perceptions of drinking pre-NF and post-NF reactions were calculated. Mediation analyses were performed using the PROCESS procedure (*Hayes*, *2013*) in SPSS 22.0 (*IBM Corp.*, *2013*). Mediation was formally tested using the product of coefficients and bias-corrected bootstrap confidence intervals (*Hayes*, *2013*), with 10,000 samples specified. Mediators were first tested independently, and effect sizes of the indirect effects were calculated using Preacher and Kelley's kappa-squared (*Preacher & Kelley*, *2011*), where appropriate. In this context, Kappa-squared provides information about the size of the indirect effect relative to its maximum possible value, given the natural constraints imposed by variances and correlations between variables (*Hayes*, *2013*). If more than one independent mediator was significant, then multiple mediation was tested. Procedures were reviewed and granted approval by the Institutional Review Board at the New York State Psychiatric Institute.

## RESULTS

### Sample description

Participants ($N = 87$) ranged in age from 18 to 64 ($M = 33$, $SD = 11$). Sixty-four percent were male.

| Table 1 | Pre-feedback descriptives of participants. |
|---|---|
| | M (SD) or % |
| **Males (N = 56)** | |
| Alcohol use is excessive[a] | 1.9 (1.5) |
| Drinking is a problem[a] | 1.6 (1.3) |
| My drinking is a problem for others[a] | 1.9 (1.6) |
| Effort needed to refuse a drink[a] | 2.6 (1.7) |
| Number of days drink per week | 2.2 (2.0) |
| Drinks per drinking day | 2.9 (2.5) |
| Binge drank in the last 30 days[b] | 62% |
| Mean standard drinks per week | 8.2 (14.2) |
| **Females (N = 31)** | |
| Alcohol use is excessive[a] | 2.0 (1.9) |
| Drinking is a problem[a] | 1.8 (1.8) |
| My drinking is a problem for others[a] | 1.8 (1.5) |
| Effort needed to refuse a drink[a] | 2.8 (2.4) |
| Number of days drink per week | 1.8 (1.5) |
| Drinks per drinking day | 2.8 (2.4) |
| Binge drank in the last 30 days[b] | 60% |
| Mean standard drinks per week | 7.9 (12.3) |
| **Severity of discrepancy[c] (N = 87)** | 42.9 (21.7) |

Notes.
[a] Range is 1 Not at all to 8 Extremely.
[b] Defined as drinking more than 5 standard drinks for men and 4 standard drinks for women.
[c] The actual percentile minus the estimated percentile.

## Descriptives

Table 1 shows the reported drinking levels for the sample. Mean number of standard drinks per week for the entire sample was 8.1 (SD = 13.1), with a range from 1 standard drink to as high as 105 for men and 60 to women. While the majority of participants were light to moderate drinkers, many participants reported drinking all of their drinks on only one or two days.

Based *only* on number of standard drinks per week, 14.9% reported drinking beyond safety guidelines set by the National Institute of Alcohol Abuse and Alcoholism for drinks per week among both men (>14 standard drinks per week) and women (>7 standard drinks per week).

### Perceptions of drinking pre-NF

There were no gender differences on any of the perception variables. Table 1 demonstrates how participants viewed their drinking. Overall, drinkers did not view their drinking as excessive, a problem, or a problem for others. Refusing a drink required little effort. Drinkers also tended to largely underestimate their drinking compared to US adults, as demonstrated by a high mean *severity of discrepancy* score.

| | (N = 87) M (SD) or % |
|---|---|
| **Table 2  Post-feedback descriptives: mediators and outcome.** | |
| **Personal dissonance variables** | |
| Surprise | 6.1 (2.2) |
| Worry | 2.6 (2.1) |
| **Expectation.** This information was: | |
| Better than expected | 2.3 |
| As expected | 10.3 |
| Worse than expected | 87.4 |
| **Yes, this information was new.** | 88.5 |
| **Belief the feedback was accurate[a]** | −.54 (1.1) |
| **Planning for change** | |
| I have no plan to change my drinking. | 73.6 |
| I plan to cut back on my drinking a little bit. | 14.9 |
| I plan to greatly cut back my drinking. | 9.2 |
| I plan to quit drinking. | 2.3 |
| **Confidence to change drinking** (n = 23)[b] | 6.3 (1.3) |
| **Commitment to change drinking** (n = 23)[b] | 5.8 (1.7) |

Notes.
[a] Range was from –2 (not at all accurate) to 0 (not sure) to 2 (definitely accurate).
[b] Range was from 1 Not at all to 8 Extremely.

### Reactions to NF

Reactions to NF are shown in Table 2. While participants were generally surprised by feedback, they were not particularly worried by it. Only 12.6% of participants viewed feedback as better than expected or as expected. A preponderance of drinkers (87.4%) reported feedback was worse than expected. A majority of drinkers (88%) reported this information was new. On average, drinkers reported that the feedback was somewhat inaccurate.

### Plan, confidence, and commitment to change

Table 2 also shows the outcome of NF. Among this sample of drinkers, 26.4% planned to quit or reduce their drinking. Confidence and commitment to change drinking were high.

## Correlations between perception of drinking pre-NF and post-NF reactions

Correlation coefficients between pre-NF perceptions of drinking and the mediators were generated (see Table 3). *Drinking is a problem* was significantly negatively correlated with *surprise* and significantly positively correlated with *worry*, *belief in the accuracy of feedback*, and *effort*. *Drinking is a problem* was also significantly negatively correlated with *severity of discrepancy*. Findings suggest when an individual viewed his or her alcohol use as problematic, there was higher accuracy in his or her estimate of how his or her drinking compared to the norm. As a result, feedback was believable and not a surprise.
**Table 3  Correlations between pre- and post-NF variables.**

| | 1 | 2 | 3 | 4 | 5 | 6 | 7 | 8 |
|---|---|---|---|---|---|---|---|---|
| | | | | | Pre-NF variables | | | |
| 1. Drinking is excessive | – | | | | | | | |
| 2. Drinking is a problem | .92*** | – | | | | | | |
| 3. Drinking is a problem for others | .47*** | .51*** | – | | | | | |
| 4. Effort | .49*** | .46*** | .33*** | – | | | | |
| 5. Severity of discrepancy | −.26* | −.29** | −.18 | −.16 | – | | | |
| | | | | | Post-NF Variables | | | |
| 6. Surprise | −.15 | −.39*** | −.15 | −.14 | .55*** | – | | |
| 7. Worry | .28** | .47*** | −.15 | .28** | .03 | −.07 | – | |
| 8. Belief in the accuracy of NF | .20 | .34** | −.15 | −.20 | −.37*** | −.60*** | .18 | – |
| 9. Plan for change[a] | .42*** | .47*** | .26* | .21 | −.10 | −.21 | .58*** | .32** |

Notes.

*$p < .05$.

**$p < .01$.

***$p < .001$.

[a]Ranged from 0 to 3, with 0 indicating no plan for change and 3 indicating quitting drinking.

NF,  normative feedback.

*Drinking is a problem* prior to NF was significantly and strongly correlated with *plan for change* (Table 3), such that the greater the participant ranked *drinking is a problem*, the more ambitious the plan for reduced drinking.

## Total effect of severity of discrepancy on plan for change

There was no total effect of *severity of discrepancy* on *plan for change* ($b = .003$, $SE = .01$, $p = .79$).

## Mediation models for personal dissonance (worry and surprise)

Table 4 displays the relationships of the mediation models. There were no significant indirect effects for either *worry* or *surprise* on *plan for change*. For this sample, personal dissonance did not mediate the effect of severity of discrepancy on making a plan for change.

## Mediation models for belief in the accuracy of NF

Table 4 also shows the results from the mediation model for *belief in the accuracy of feedback*. *Severity of discrepancy* was significantly negatively associated with *belief in the accuracy of feedback*. There was a significant indirect effect for *belief in the accuracy of feedback* (.004, 95% CI [.001–.01]). The kappa-squared statistic revealed that the indirect effect is 11.6% of its maximum possible value ($K^2 = .116$, $SE = .06$, 95% CI [.03–.25]).

## Post hoc analyses

Since this was a sample of mostly light to moderate drinkers, rather than predominantly hazardous drinkers, it was surprising to discover such a large proportion of them (26.4%) reported planning at least some change in their drinking. Due to the absence of a relationship between *plan for change* and *severity of discrepancy*, it is possible that other baseline variables impact *plan for change*. Given the relationship between pre-NF *drinking is a problem* to

**Table 4 Mediation models for worry, surprise, and perceived accuracy of feedback predicting plan to change drinking.**

| | | | | | | | | | |
|---|---|---|---|---|---|---|---|---|---|
| | | | | Outcome variable | | | | | |
| | | Mediator | | | | Plan to change drinking | | | |
| | | Coeff | SE | P | | | Coeff | SE | P |
| **Model 1** | | | | | | | | | |
| X (Severity of discrepancy) | $a$ | .003 | .01 | .79 | $c'$ | | .004 | .003 | .19 |
| M (Worry) | | – | – | – | $b$ | | .21 | .03 | <.001 |
| | | $R^2 = .001$ | | | | | $R^2 = 0.35$ | | |
| | | $F(1, 85) = 0.07, p = .79$ | | | | | $F(2, 84) = 22.8, p < .001$ | | |
| **Model 2** | | | | | | | | | |
| X (Severity of discrepancy) | $a$ | .06 | .01 | <.001 | $c'$ | | −.001 | .01 | .82 |
| M (surprise) | | – | – | – | $b$ | | .08 | .05 | .09 |
| | | $R^2 = .30$ | | | | | $R^2 = 0.04$ | | |
| | | $F(1, 84) = 35.9, p < .001$ | | | | | $F(2, 83) = 1.9, p = .16$ | | |
| **Model 3** | | | | | | | | | |
| X (Severity of discrepancy) | $a$ | −.02 | .01 | <.001 | $c'$ | | −.001 | .004 | .84 |
| M (Perceived accuracy) | | – | – | – | $b$ | | −.22 | .07 | <.01 |
| | | $R^2 = .13$ | | | | | $R^2 = 0.11$ | | |
| | | $F(1, 85) = 13.0, p < .001$ | | | | | $F(2, 84) = 4.9, p < .001$ | | |

post-NF *plan for change*, post hoc analyses were implemented to further explore this relationship. *Drinking is a problem* was independently tested as a predictor of *plan for change* using stepwise linear regression.

Pre-NF *drinking is a problem* significantly predicted *plan to change* ($b = .24$, $SE = .05$, $p < .001$). When post-NF *worry* was entered as a second step in the model, both remained significant (*drinking is a problem*: $b = .24$, $SE = .05$, $p < .001$; *worry*: $b = .24$, $SE = .05$, $p < .001$; adjusted $R^2 = .38$; $F(2, 84) = 26.8$, $p < .001$), suggesting that both baseline perception that *drinking is a problem* and post-NF *worry* predict *plan for change*. Tolerance statistics demonstrated no multicollinear effect.

## DISCUSSION

This is the first study to test both personal dissonance and perceived accuracy of feedback as mechanisms of action within NF among adult drinkers outside the context of MET. Hypotheses that severity of discrepancy would be mediated by personal dissonance and belief in the accuracy of feedback were only partially supported. While the participants in this study were primarily light to moderate drinkers, a number of them greatly underestimated their level of drinking compared to their same gendered peers, demonstrating a wide range of discrepancy. Still, *severity of discrepancy* was not associated with *plan to change*, which is inconsistent with previous studies on college students (*Walters et al., 2009*; *Walters, Vader & Harris, 2007*). Interestingly, findings were consistent with previous research on a sample of hazardous drinkers from the general population (*Cunningham, Murphy & Hendershot, 2015*). This recent study found that norm comparisons, provided
in the context of more detailed personalized feedback, did not appear to be the active mechanism for change among a sample of adult hazardous drinkers (*Cunningham, Murphy & Hendershot, 2015*). It is possible that peer comparisons work especially well for individuals in their late adolescence and emerging adulthood because it leverages their particular need for conforming with peers' behaviors. Given the lack of association between *severity of discrepancy* and *plan for change*, it also suggests that among a sample of mostly light to moderate drinkers, factual normative discrepancy does not act alone to instigate thoughts about change.

While there were few heavy drinkers in this sample, lack of findings regarding personal dissonance was consistent with previous studies on affective responses (*Kuerbis et al., 2014*; *McNally, Palfai & Kahler, 2005*) of heavy drinkers. According to evidence thus far, it appears that affective responses are not associated with outcomes of feedback among both college-age and an age-inclusive sample of adults with a spectrum of drinking patterns.

Perceived accuracy of feedback was demonstrated to be a potential mediator of *severity of discrepancy*, providing initial support for decision dilemma theory. Results suggest that the more an individual is aware of how much he or she drinks compared to others, the more likely he or she will perceive the feedback to be accurate and thus more likely to set a plan for change. Among this sample, this was true regardless of intensity of drinking. It is important to note that the mediation effect of *belief in the accuracy of NF* was quite small—only 11.6% of its possible maximum value. More research is needed to further understand this potential mediator of change.

Post hoc analyses support the possibility that feedback for drinkers may not operate to create concern about or awareness of excessive drinking where none existed. Instead, it seems to potentially capitalize on a pre-existing perception that drinking is a problem, particularly among those who may drink more heavily. Some studies demonstrate NF works best for those with the most severe drinking patterns (*Murphy et al., 2001*; *Doumas & Andersen, 2009*). It is therefore possible that feedback works within a framework of consonance rather than dissonance. This may explain why a "boomerang effect," an effect in which light drinkers who receive feedback about their drinking increase instead of decrease their drinking (*Prince et al., 2014*), has yet to be found within the empirical literature. Furthermore, given the correlational relationships, perceiving alcohol is a problem may also indicate an awareness of one's drinking and already how it compares to existing norms.

Given that NF is one of the most widely used interventions to prevent and/or reduce hazardous drinking, the above findings have potentially important implications for application to clinical practice. When implementing NF to an adult sample, providers should be aware of both the existing concerns and perceptions a drinker may have, as well as how believable the information imparted is for the recipient. Inquiring about a patient's drinking and their perception of its role in his or her life a priori may help providers navigate providing feedback in an optimal way. For example, there may be little need to offer much additional feedback when one already perceives their drinking as problematic. Instead, offering an opportunity to explore goal planning and/or treatment may be the best use of a clinician's time. For those who do not perceive their drinking as problematic,

feedback interventions may be best improved by increasing the credibility of the feedback for the individual. By being selective in how one provides feedback, providers may enhance its effectiveness.

## Limitations

There are several limitations to this study, and findings should be interpreted in the context of an exploratory pilot study. While this study demonstrated the feasibility of measuring affective responses and *belief in the accuracy of feedback* post-NF, generalizability is extremely limited. These participants were recruited from an online labor market, not the general population. The measurement of *belief in the accuracy of feedback* was simple in that it was a single item, self-report measure; however, it had face validity in that it captured the participant's perception rather than behavior, which was the specific construct of interest. A major limitation of this study is that there were few individuals who were heavy or hazardous drinkers, in terms of quantity of standard drinks per week, and thus results cannot be generalized to hazardous drinkers. More research on understanding this relationship with hazardous drinkers is needed; however, the greater variability in the sample allowed us to understand feedback in a heterogeneous drinking sample, such as those found in primary care or other opportunistic settings, in which NF is often used preventively.

Drinking was entirely self-report and could not be corroborated by other sources. Outcomes were based on intention to change health behaviors, not actual behavior change. While other studies have used a similar outcome, it is not equivalent to behavior. Finally, it is not known whether personal dissonance is experienced only in a proximal fashion. For example, a person who receives feedback may not initially experience worry but may find his or her concern grows over a period of time. This potential sleeper effect of NF could in turn potentially affect the timing of making a plan for change. Longitudinal research is needed to better understand the timing of potential mediators and subsequent behavioral outcomes.

## Future research

Future research could include experimental manipulation with better operationalization of normative feedback, self-ideal feedback and emotional responses in order to unpack the active ingredients of NF. Expanding the population upon which NF is tested to heavy drinkers and other undesirable health behaviors will be crucial to confirming these preliminary findings. Additionally, both actual behavioral outcomes and longer term, prospective perceptions of feedback must be explored to understand the mechanisms of maintenance of behavior change.

## Conclusions

Despite the limitations, findings have the potential for immediate clinical application, and thus implications for individual and population level health. For individuals with a skewed perception of how much they drink compared to their peers, helping them to understand NF in context and making it believable (e.g., supported by empirical data) may be critical for maximum effectiveness of NF among adults. In addition, exploring with drinkers how

much they are already perceive their drinking to be a problem may help to potentiate the effects of the simplest form of NF. These are relatively minor and feasible changes to NF that may in fact help to increase its efficacy rate, potentially helping to reduce the prevalence rate of hazardous alcohol use or prevent the development of AUD across a wide group of drinkers.

### Funding

This work was funded by the National Institute on Alcohol Abuse and Alcoholism, in the United States, Grant R34 AA 021502A (PI: Muench). The funders had no role in study design, data collection and analysis, decision to publish, or preparation of the manuscript.

### Grant Disclosures

The following grant information was disclosed by the authors:
National Institute on Alcohol Abuse and Alcoholism: R34 AA 021502A.

### Competing Interests

Frederick J. Muench is an Academic Editor for PeerJ.

### Author Contributions

- Alexis Kuerbis conceived and designed the experiments, performed the experiments, analyzed the data, contributed reagents/materials/analysis tools, wrote the paper, prepared figures and/or tables, reviewed drafts of the paper.
- Frederick J. Muench conceived and designed the experiments, performed the experiments, analyzed the data, contributed reagents/materials/analysis tools, wrote the paper, reviewed drafts of the paper.
- Rufina Lee and Juan Pena analyzed the data, wrote the paper, reviewed drafts of the paper.
- Lisa Hail performed the experiments, wrote the paper, reviewed drafts of the paper.

### Human Ethics

The following information was supplied relating to ethical approvals (i.e., approving body and any reference numbers):

Hunter College IRB reviewed this project. It was deemed exempt for human subjects research because no identifying information was collected about our subjects.

### Data Availability

The raw data has been supplied as a Supplemental Dataset.

### Supplemental Information

Supplemental information for this article can be found online at http://dx.doi.org/10.7717/peerj.2114#supplemental-information.

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
