# Peer review of "An exploratory pilot study of mechanisms of action within normative feedback for adult drinkers"

_PeerJ, doi:10.7717/peerj.2114_

## Round 0.1 · original submission · Major Revisions

Please address the reviewer comments. You need to develop your introduction to include a clear description of Normative feedback, and include examples of feedback provided as part of your methodology. Your paper requires thorough proof reading.

Reviewer 1 ·

Basic reporting

1. Given that the authors are reporting on NF, a specific type of brief intervention, a brief description of what brief interventions are would be useful for the reader unfamiliar with them. Indeed, previous authors have called for attention to mechanisms of brief interventions as this is lacking in the literature, so this literature would support this work.
2. The authors seem to underplay the extensive literature on brief interventions and normative feedback – they should highlight recent syntheses from this field and findings across these syntheses (e.g., A Corrective Meta-Analysis of Personalized Normative Feedback by David Zimmermana & Baruch Fischhoff; Personalized Drinking Feedback: A Meta-Analysis of In-Person versus Computer-Delivered Interventions by Cardigan et al., 2015)
3. The authors state that the perceived accuracy measurement of a previous study was limited (line 112), but give no indication about what its limits were. Their own measure of perceived accuracy was also quite simple (although I’m not aware of the literature base for how best to measure perceived accuracy). Additional details would be useful on how theirs was an improvement.
4. Line 119: “In this study, personal dissonance” should be followed by “defined as surprise or worry”
5. One or two sentences on why the kappa-squared is useful – how does it help in interpretation- should be added to the analytic plan section.

Experimental design

1. Although the IRB approval is reported, it seems the authors should include one line in the text of the article that IRB approval was granted.
2. How many participants answered “no” to the question about drinking alcohol?
3. Regarding the normative feedback given – was there only one statement of normative feedback? If more than one, then it would be useful to have all statements listed somewhere as this is a report of an exploratory feasibility study and would be useful to others interested in this approach and provide transparency around the method.
4. Lines 216-230 should be expanded and the table listing the numeric values summarized should be edited. The table is quite confusing and the authors seem to leave out the “c” pathway completely, focusing only on c1. I would suggest use of a path diagram with the coefficients listed on each path and c1(c) listed as well. Path diagrams can be much easier to interpret. Specifically, the text should include more of the numeric details and significant pathways should be reinterpreted in light of the measures used. For example, is a significant indirect effect of 0.004 for perceived accuracy meaningful in light of the measures used?
5. Line 253 discusses “change in worry”, but I did not see where worry was measured both pre- and post- NF. If so, this should be added to the methods. Or this statement should be reworded for clarity.
6. Table 4 was confusing because I don’t see the “plan for change” variable anywhere. It looks like a correlation matrix, but missing the key outcome variable. It also seems that the correlations should be reported prior to running the mediation models.

Validity of the findings

The authors clearly linked results to their conclusions and stated their limitations. It seems problematic that the only number reported in the text of the results (prior to post hoc analyses) was the one significant positive effect, leading readers to focus on that one. I would suggest that the authors restructure this section substantially. Although I expect that some conclusions may change if they revisit their results section, I do not have further comments.

Additional comments

1. It seems that in the abstract the conclusions should be more carefully stated, e.g., “Alcohol feedback for adult light drinkers”, rather than a broad statement about all types of alcohol feedback.
2. I would suggest that line 232 be slightly reworded to avoid stigmatizing language. For example: “Given that this was a sample of self-reported light drinkers rather than those that report problematic drinking behaviors…”
3. Table 2: “This information was new” should be on its own line because otherwise it looks like it is part of the other categorical variable (“This information was better, as, worse than expected”)

Reviewer 2 ·

Basic reporting

Introduction & Scope of Journal: Overall, this paper seems to be geared toward social sciences, though it is clear that there are health science applications for NF. The introduction should include a paragraph explaining how NF for drinking fits into the larger context of health sciences. Specifically, you may consider expanding on your sentence on lines 57-58 to further discuss the potential of NF as a prevention/early intervention approach for reducing unhealthy alcohol use in primary care and how efforts to validate population-level interventions are consistent with the aims of healthcare reform. See the following articles for a discussion of population-level public health considerations for NF:
1. Dotson, K. B., Dunn, M. E., & Bowers, C. A. (2015). Stand-Alone Personalized Normative Feedback for College Student Drinkers: A Meta-Analytic Review, 2004 to 2014. Plos ONE, 10(10), 1-17. doi:10.1371/journal.pone.0139518
2. Bertholet, N., Cunningham, J. A., Faouzi, M., Gaume, J., Gmel, G., Burnand, B., & Daeppen, J. (2015). Internet-Based Brief Intervention to Prevent Unhealthy Alcohol Use among Young Men: A Randomized Controlled Trial. Plos ONE, 10(12), 1-11. doi:10.1371/journal.pone.0144146

Writing/Language: Throughout the manuscript the writing is unclear and in some sections incoherent and interferes with readability and comprehension of the content. The authors should edit the manuscript to provide concise language that does not detract from the content. For example, outcome variables should be identified in brief, clear terms. See annotated PDF for specific recommendations.

Experimental design

Reproducible methods: Authors should provide a sample or a more detailed description of the feedback that was provided to participants to allow for replication by other researchers and to compare the NF intervention content to that used in previous studies.

See annotated PDF for comments.

Validity of the findings

No comments.

Annotated reviews are not available for download in order to protect the identity of reviewers who chose to remain anonymous.

---

## Round 0.2 · accepted · Accept

Please address the minor typological issues identified by the reviewer in the production phase.

Reviewer 1 ·

Basic reporting

The authors have made a substantial number of changes and addressed all previously raised concerns. The figures, especially table 4, and the discussion of results are much easier to follow.

Experimental design

No comments

Validity of the findings

No comments

Additional comments

Overall, I do not have any major suggestions for the article.

There are a few places with small typos to address:
Line 160 - "outside of the session" (add "the")
Line 232-233 - "-2 (Not at all accurate) to 2 (Definitely
Manuscript to be reviewed)" (add the "()"
Line 260-261 - I would move this line to be earlier in the methods section, or at least prior to the analysis section: "Procedures were reviewed and granted approval by the Institutional Review Board at the New York State Psychiatric Institute."
Line 418 - "In addition, exploring with drinkers how much they
already" (remove "are")

Reviewer 2 ·

Basic reporting

The authors' copy edits significantly improved overall readability and flow of the manuscript. Content edits helped to place this study in the larger context of the field.

Experimental design

The methods section was much improved by the addition of detailed procedural description.

Validity of the findings

No comments

Additional comments

The authors were clearly responsive to this reviewer's feedback and made edits consistent with previously stated recommendations. The manuscript is much improved and would be a useful contribution to the field. It is my recommendation that this manuscript be accepted for publication.